# Delayed Bronchopleural Fistula Formation Following Salvage Surgery of Stage IV Anaplastic Lymphoma Kinase-Positive Non-Small-Cell Lung Cancer

**DOI:** 10.3390/curroncol32050250

**Published:** 2025-04-25

**Authors:** Lauren Barter, Stephanie Snow, Aneil Mujoomdar, Lara Best, Daniel French

**Affiliations:** 1Faculty of Medicine, Carleton Campus, Dalhousie University, 5849 University Ave., Halifax, NS B3H 4R2, Canadalara.best@nshealth.ca (L.B.); 2Division of Medical Oncology, Department of Internal Medicine, Halifax, NS B3H 2Y9, Canada; 3Department of Thoracic Surgery, Victoria Building, Room 7-014, 1276 South Park Street, Halifax, NS B3H 2Y9, Canada; 4Department of Radiation Oncology, Dickson Building, Room 2200, 5820 University Avenue Halifax, Halifax, NS B3H 1V7, Canada

**Keywords:** lung cancer, ALK targeted therapy, salvage surgery, bronchopleural fistula, serratus muscle flap

## Abstract

This case report highlights the management of a delayed bronchopleural fistula (BPF) following salvage pulmonary resection to achieve local control and no radiographic evidence of disease in a patient treated with serial tyrosine kinase inhibitors (TKIs) for stage IV anaplastic lymphoma kinase (ALK)-positive non-small-cell lung cancer (NSCLC). The initial pulmonary resection was complicated by dense adhesions and an abnormally torturous pulmonary artery. Six weeks after the index surgery, the patient presented with a delayed BPF requiring decortication, repair of airway, and coverage of the bronchial stump with a serratus anterior muscle flap.

## 1. Introduction

Anaplastic lymphoma kinase (ALK)-positive lung cancer is relatively rare and is prevalent among those with minimal or no tobacco exposure. As a result of constitutive activation of the ALK kinase, ALK-positive non-small-cell lung cancer (NSCLC) exhibits dysregulated cellular proliferation and survival through downstream signaling pathways [1]. This alteration, found in approximately 3–7 percent of advanced NSCLC cases, is often associated with younger age, non-smoking status, and adenocarcinoma histology [2].

Targeted therapies, particularly sequential TKIs, have transformed the management of ALK-positive lung cancer, significantly prolonging patient survival. Over the past two decades, the advent of personalized therapies has revolutionized the management of advanced NSCLC by targeting specific oncogenic drivers, which have markedly improved response rates, survival, and quality of life for molecularly selected patients [3]. The integration of small-molecule ALK inhibitors has significantly advanced treatment options for ALK-positive cancers. However, clinical challenges remain in situations such as oligoprogression, where emerging resistant lesions accompany otherwise locally controlled disease [3]. This condition necessitates a nuanced approach that includes next-generation ALK inhibitors and potential local therapies including radiotherapy, radiofrequency, and surgery [4]. When assessing which targeted therapies are appropriate, clonal evolution and the emergence of resistance mutations can play crucial roles in treatment decisions; however, the resources to test for these is often not available in standard of care practice. Ongoing utilization of targeted therapies, coupled with personalized local treatment strategies, is crucial for managing the complexities of oligoprogression and achieving optimal long-term outcomes [5,6,7].

Stage IV ALK-positive non-small-cell lung cancer (NSCLC) presents a challenge regarding treatment options and long-term management. The emergence of targeted TKIs has changed the landscape for patients with this disease, with many having long periods of disease response. For those initially treated with earlier-generation ALK TKIs, we know that treatment with serial ALK TKIs, when available, is the optimal standard of care for systemic therapy for this patient group [8]. Salvage surgery is considered in select cases with a good response to systemic therapy, aiming to provide localized control and potentially improve outcomes. Aggressive local ablative therapy (LAT) plays a crucial role in oligoprogressive disease, where progression occurs at limited sites, while systemic therapy maintains control elsewhere at sites that have maintained systemic treatment sensitivity. LAT options include surgery, radiation therapy, radiofrequency ablation, or cryoablation for these progressing sites while continuing systemic treatment [9].

The occurrence of a bronchopleural fistula (BPF) following lobectomy is a rare complication that demands prompt recognition and comprehensive management. BPF is a condition that occurs when there is an abnormal connection between the airway of the lung and the pleural space. Although it can occur at any time, it is most frequently observed within the first 8 to 12 days after the procedure. It is important to be aware of this potential complication and to monitor patients carefully during this critical time to ensure early detection and appropriate intervention [10].

## 2. Case Report

A 53-year-old women with no history of tobacco use presented with cough and dyspnea during the autumn of 2016. A chest X-ray on 17 February 2017 showed a well-circumscribed spiculated right middle lobe (RML) lung mass adjacent to the right hilum. Follow-up CT on 1 March 2017 confirmed a 4.5 cm × 3.1 cm spiculated RML lesion, which was abutting and crossed the major fissure, bilateral mediastinal lymphadenopathy, and satellite nodules throughout the right lung, as shown in Figure 1. A positron emission tomography (PET) CT on 31 March 2017 showed intense fludeoxyglucose-18 (FDG) avidity of the RML mass with an SUV max of 20.5, in addition to FDG avid right hilar, subcarinal, paratracheal, and extrathoracic right subclavicular lymph nodes (Figure 2). Finally, there was an FDG avid pleural-based nodular density noted in the low anterior right costophrenic angle region (M1a).

Video-assisted thoracoscopic surgery (VATS) was undertaken on 3 April 2017, during which the FDG avid pleural-based lesion was biopsied. Pathologic assessment confirmed a poorly differentiated adenocarcinoma that tested positive for an ALK fusion using the 5A4 immunohistochemical clone that was standard of care for ALK testing at that time. Molecular genotyping using a next-generation sequencing platform that did not include ALK revealed no gene mutations, including KRAS, EGFR, BRAF, AKT1, HER-2, NRAS, PIK3CA, and MET. The initial clinical stage was T2 N3 M1A (AJCC 8th edition clinical stage IVA).

She started first line systemic therapy with crizotinib 250 mg orally twice a day on 25 April 2017 for 2.5 years. In August of 2018, she received 60 Gy in eight fractions of stereotactic body radiotherapy (SBRT) to the primary RML lobe tumor, which had exhibited progression on crizotinib. As this was the only site of active disease, stereotactic radiation therapy to the right middle lobe lesion was considered as the method to control this area. Additionally, it provided the option to stay on crizotinib for a longer period of time.

A surveillance computed tomography (CT) scan 28 months after beginning crizotinib treatment revealed further progression in the primary RML mass and a hilar lymph node. Second-line alectinib 600 mg twice daily was started on 1 November 2019 in the context of enrollment in the ALTA-3 phase three clinical trial [11]. Ten months later, CT surveillance revealed the tumor met RECEIST criteria for disease progression, and third-line treatment with lorlatinib was initiated and has been ongoing since 14 August 2020.

Five and a half years after initiating systemic therapy, a surveillance CT on 26 October 2022 demonstrated progression at the site of the primary tumor. A PET scan on 2 February 2023 suggested the pleural metastatic and nodal disease had undergone a complete radiographic response but showed activity at the initial primary site (Figure 3). After review by a multidisciplinary tumor board, a decision was made to proceed with anatomical resection.

A right posterolateral thoracotomy sparing the anterior serratus but dividing the latissimus dorsi was performed. On entry into the thorax, dense adhesions were encountered between the lung and chest wall due to post-radiation fibrosis and pleural wall thickening. Dissection in the hilum and fissure revealed that, despite the tumor originating in the RML, it crossed the fissure, and therefore a decision was made to proceed with an RLL/RML bilobectomy. An air leak test using saline to submerge the bronchial stump was performed, confirming the bronchial closure was air-tight with horizontal mattress polydioxanone sutures. Thymic fat was used to buttress the bronchial stump. The total blood loss from the surgery was 4500 cc, which resulted from the decortication and pulmonary artery injury. The patient received five units of packed red blood cells during the operation. Lorantinib was restarted on POD 1 and was not held perioperatively.

The early postoperative course was complicated by a pleural effusion requiring re-insertion of a chest drain on postoperative day (POD) 8. Her discharge home was delayed by poor mobility, but she was ultimately discharged home on POD 13. Six weeks after discharge, she presented with dyspnea and cough and was found to have a pinhole pulmonary bronchopleural fistula on bronchoscopy and empyema on CT. Cultures from the empyema were positive for nocardiosis and aspergillosis. Antifungal and antibiotic treatment with voriconazole and sulfamethoxazole/trimethoprim was initiated, which necessitated a reduction in the dosage of loratinib to 75 mg daily.

The patient returned to the operating room 62 days after the initial operation to re-open the initial thoracotomy, taking care to protect the serratus anterior on entry into the chest, and decortication of the right hemithorax and repair of the bronchus were performed. At the time of opening the thoracotomy, gelatinous material was present in the pleural space, suggesting an empyema. After decortication, a dehiscence of the bronchus intermedius bronchial stump was obvious. This was debrided and closed with horizontal mattress sutures and buttressed with an anterior serratus muscle flap. The patient recovered uneventfully from the second surgery and was discharged home on POD 15.

The final pathology revealed complete resection of a ypT2bN0 adenocarcinoma. The tumor measured 4.3 cm with mixed histology, including 70 % acinar, 25% solid, and 5% micropapillary components. There was no associated lymphovascular invasion. It was positive for the known EML4(13):ALK(20) fusion (380 reads). Next-generation sequencing revealed a new MET mutation (H1094Y) with an 8.3% variant allele frequency and ALK mutation V1135L with a 9.1% variant allele frequency.

The most recent restaging CT performed on 20 January 2025 showed no evidence of active recurrent or metastatic disease and stable post-surgical changes. She is tolerating lolatinib well with a dose of 75 mg daily in the context of ongoing antimicrobial therapy. She does have grade 2 dyspnea on exertion but is feeling better in general.

## 3. Discussion

The present case provides three noteworthy observations: (1) a patient with oligoprogressive ALK-positive lung cancer managed aggressively with targeted therapy, SBRT, and surgery had no radiographic evidence of disease more than seven years from diagnosis of stage IV lung cancer, (2) salvage resection was undertaken albeit with challenges in dissection, and (3) the patient subsequently developed a delayed BPF.

Numerous case series have extensively documented the possibility of performing salvage surgery following chemotherapy and radiation [12,13,14,15]. These cases typically involve surgery after chemotherapy and/or radiation, where the original tumor was previously considered unresectable due to locally advanced disease [16]. However, there are few documented cases of patients transitioning from stage IV to localized disease exclusively [17,18]. Additionally, there are limited data on the role of surgery after targeted therapy [19,20,21,22,23,24]. In this particular case, the patient’s initial diagnosis was stage IVA NSCLC following a biopsy of the parietal pleura. However, after undergoing three lines of systemic therapy that targeted ALK, as well as stereotactic body radiation therapy (SBRT) directed at the primary tumor, she had no radiographic evidence of disease post-resection of primary tumor.

The resected tumor had molecular profiling with evidence of on- and off-target mutations, presumably the mechanisms of resistance to the most recent lorlatinib TKI therapy. MET mutations have become a significant mechanism of resistance to ALK inhibitors in ALK-rearranged non-small-cell lung cancer. Research has demonstrated that MET activation can contribute to resistance by reactivating downstream signaling pathways. Current efforts are focused on exploring combination approaches that involve MET inhibitors to address this resistance mechanism. However, there is limited clinical experience with the combined inhibition of ALK and MET so far [25].

When patients require prolonged management of systemic disease, it is crucial to explore local control options in the setting of oligopression. Multidisciplinary review of radiological and pathological findings is integral to making the best treatment plan for each patient, including aggressive local therapy. Real-world evidence generation using population data will be integral to defining the impact of such strategies on long-term survival.

When considering resection, it is important to anticipate the potential for arduous and lengthy surgeries, as was the case here, where two surgeons were required for over 6.5 h. The presence of dense adhesions due to prior VATS and radiation, as well as potential tumor response to systemic therapies, can compound the difficulty of the procedure. Notably, during hilar dissection, pulmonary arteries were found to be atypical, tortuous, and deformed, possibly due to tumor reaction to targeted therapy and radiation-induced anatomical changes. Careful preoperative planning, including the involvement of a second surgeon to assist with technical maneuvers and decision making during surgery, is essential. Advanced techniques such as bronchoplasty and angioplasty, along with the proximal and distal control of the pulmonary arteries and veins, are indispensable. Additionally, it is crucial to anticipate and prepare for postoperative complications that may arise from the intricacies of dissection and reconstruction following prolonged operations.

While salvage surgery in advanced NSCLC treated with targeted therapy may be an effective tool to control sites of oligoprogression in patients with oncogene-addicted stage IV lung cancer, the emergence of rare complications such as bronchopleural fistula necessitates vigilance and strategic intervention. Holistic management involving a multidisciplinary team remains pivotal in optimizing outcomes in such scenarios.

The present case highlights a noteworthy occurrence of a rare complication: a delayed bronchopleural fistula (BPF) following anatomical resection. Diagnosis was confirmed six weeks and five days postoperatively, with symptoms emerging earlier, albeit not within the initial two-week post-resection period, during which the patient remained hospitalized.

BPF arises from the failure of bronchial stump closure after surgery, attributed to four primary etiological factors: technical inadequacy (evidenced as incomplete closure at the index operation), extensive tension (encountered during anastomosis in sleeve resections), local contamination with bacteria generating proteases leading to tissue breakdown, or inadequate blood supply.

In this instance, the bronchopleural fistula was likely attributable to ischemia secondary to radiation-induced changes from stereotactic radiation administered three and a half years preceding the resection. However, the impact of surgery and targeted therapies on wound healing, coupled with local malignant cell responses, remains incompletely understood. When managing late BPF, various factors influence intervention choices, such as the patient’s clinical stability and the anatomical characteristics of the lung and pleural space. In this case, the patient was deemed clinically stable, and there was adequate residual lung in the right hemithorax to fill the pleural space. Options like lung decortication and direct fistula repair were considered. The possibility of a two-stage procedure, such as the Clagett procedure, which initially drains the infected cavity and allows time for the fistula to mature before definitive closure, was also explored. A two-stage approach would likely have been favored if the patient was not expected to withstand a more aggressive single-stage procedure. Ultimately, primary closure with a muscle flap buttress was performed, given the patient’s overall condition and the feasibility of lung decortication.

In this case, the airway closure was meticulously performed with horizontal mattress sutures, ensuring the reapproximation of the membranous and cartilaginous portions to minimize tension. A negative leak test upon airway submersion in saline with concurrent ventilation to the right lung confirmed successful repair. The repair was further reinforced with a substantial portion of pericardial fat to augment blood supply and bulk. Nonetheless, a BPF ensued despite these measures, necessitating decortication and coverage with a serratus muscle flap.

The occurrence of a delayed BPF following salvage surgery in a patient with advanced non-small-cell lung cancer (NSCLC) treated with serial ALK TKIs and lung stereotactic radiation presents a clinical conundrum. Prompt recognition and a multidisciplinary approach are indispensable in managing such complex complications. This case underscores the significance of close postoperative monitoring and considering surgical intervention in refractory cases, notwithstanding initial attempts at conservative management.

## 4. Conclusions

This report presents a case where a patient was initially diagnosed with stage IV cancer and had experienced significant survival with serial ALK TKI systemic therapy with appropriate interventional with aggressive local therapies to address oligoprogression, first with SBRT and later with surgical resection of the primary site. The report documents the effectiveness of salvage resection in such cases and emphasizes the importance of careful attention to intra-operative technical details and postoperative adverse events. As thoracic oncology continues to gain experience with targeted therapies, the role of surgery in treating oligopression will be further understood.

## Figures and Tables

**Figure 1 curroncol-32-00250-f001:**
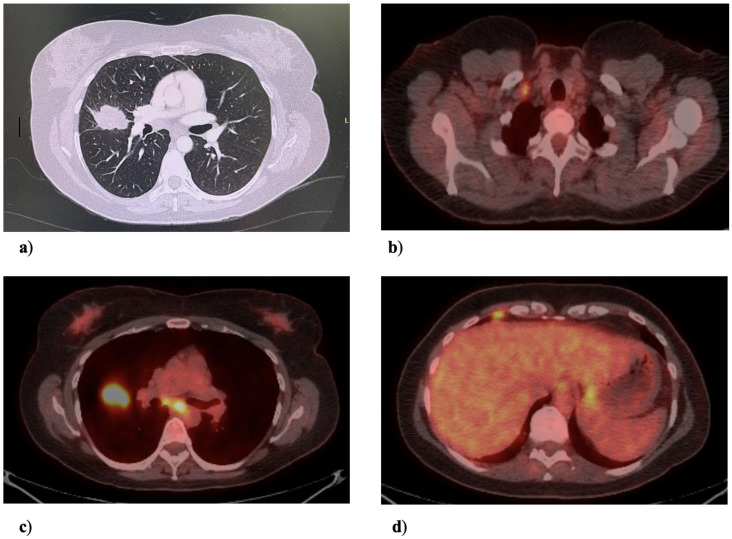
(**a**) Computed tomography revealing a 4.5 × 3.1 cm opacity in the right middle lobe lung adjacent to the right hilum. (**b**) Positron emission tomography (PET) slice uptake corresponding to the abnormal fludeoxyglucose (FDG) enhancement in a right-sided extrathoracic subclavicular lymph node at the T3 level. (**c**) PET scan slice showing FDG uptake in mediastinal lymph nodes. (**d**) PET scan slice showing FDG uptake in a pleural-based nodular density in the low anterior right costophrenic angle region.

**Figure 2 curroncol-32-00250-f002:**
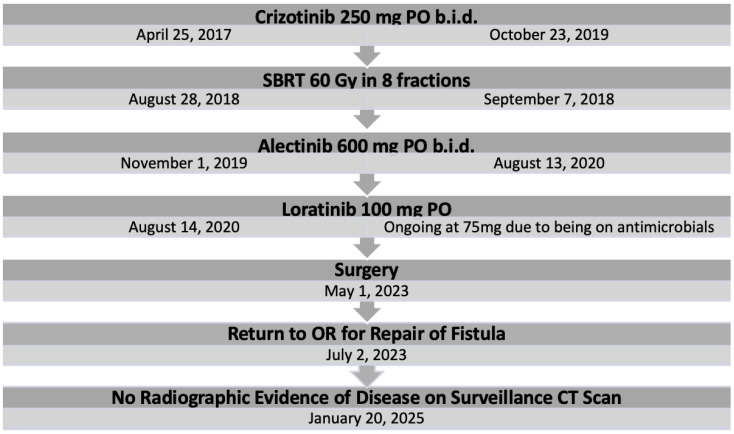
Treatment timeline.

**Figure 3 curroncol-32-00250-f003:**
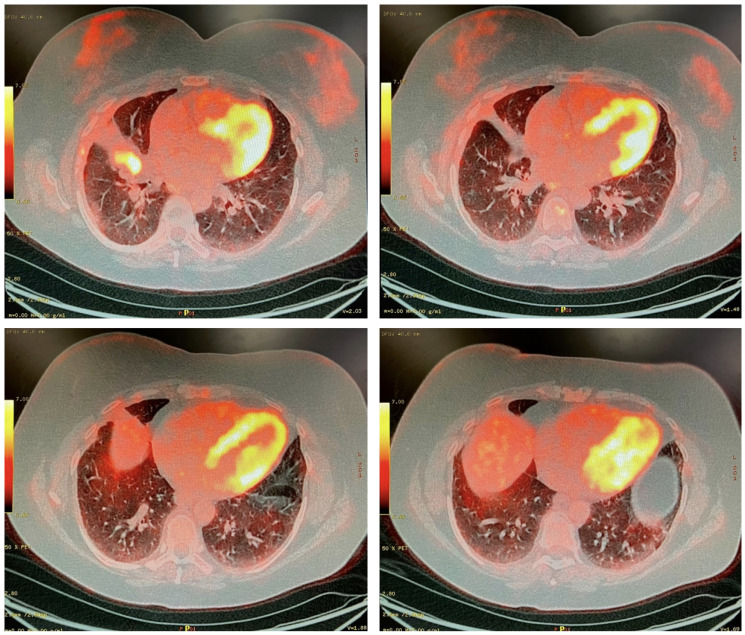
PET scan, 2 February 2023. The suspected site of recurrent disease in the right middle lobe was intensely FDG-avid. The lesion metabolically measured 3.6 × 1.9 cm. No other FDG-avid primary nodule or mass was seen. There was no FDG-avid thoracic adenopathy.

## Data Availability

Data is contained within the article.

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
