# Peer review of "Delayed Bronchopleural Fistula Formation Following Salvage Surgery of Stage IV Anaplastic Lymphoma Kinase-Positive Non-Small-Cell Lung Cancer"

_curroncol, 2025, doi:10.3390/curroncol32050250_

Round 1
Reviewer 1 Report
Comments and Suggestions for Authors
This is a well written case report of a bronchopleural fistula following a bilobectomy in a patient with a complex oncologic history. While BPF following bilobectomy is rare, it is not all that novel. This patient had a remarkable response to a third line ALK-TKI and given patient factors the authors undertook a heroic resection for local control. Given the prior treatment including SBRT the patient was at higher risk for complications following the surgery. I agree with the authors' conclusion that it was likely the prior radiation that was the primary risk factor for the BPF. However, I think it would be of interest to include a discussion if ALK-TKI's have an impact on wound healing which may have contributed to the described complication. Along this line of thinking, can the authors comment on when the patient was restarted on Loratinib post-operatively? I also would like further clarification on the following questions: how was the bronchus closed at the index operation, stapled or hand-sewn. Was there a discussion of a more robust tissue flap coverage of the bronchial stump at the time of the index operation such as serratus m. or intercostal m. Did the patient have any imaging after discharge from the index operation demonstrating fluid or a large space? How was the patient diagnosed with the BPF, bronchoscopy, CT? Did the patient have a chest tube placed on re-admission? How long were antibiotics administered prior to going to the OR for bronchial stump closure. This is a well written case report and could be of interest if these questions are addressed as we continue to treat more complicated patients in thoracic surgery.
Reviewer 2 Report
Comments and Suggestions for Authors
First of all I would like to congratulate the authors for their great work and result with the patient presented. I have some contributions that I think could improve the work.
- First, I think it would be important to provide CT and/or PET images prior to surgery (post Qt and SBRT).
- Secondly, it is not explained how the bronchopleural fistula was diagnosed. Was it by bronchoscopy? What was fistula`s size?
- Finally, how was the type of fistula intervention decided? Being an infected cavity as it is usually in late fistulas, was the possibility of performing surgery in two stages (Clagett procedure) or another option considered? I think it would be interesting to introduce in the discussion the therapeutic possibilities of late fistulas and how they decided on primary closure.
